

# First measurement results from DANAE - Demonstrating DePFET RNDR on a prototype Matrix

Alexander Bähr[1*], Holger Kluck[2], Peter Lechner[1], Jelena Ninkovic[1], Jochen Schieck[2,3], Hexi Shi[2] Wolfgang Treberspurg[2,4] and Johannes Treis [1]

**1** Halbleiterlabor der Max-Planck Gesellschaft, Otto-Hahn-Ring 6 81739 Munich, Germany
**2** Institut für Hochenergiephysik der Österreichischen Akademie der Wissenschaften, Nikolsdorfer Gasse 18, 1050 Vienna, Austria
**3** Atominstitut, Technische Universität Wien, Stadionallee 2, 1020 Vienna, Austria
**4** Fachhochschule Wiener Neustadt, Johannes Gutenberg-Straße 3, 2700 Wiener Neustadt, Austria

⋆ axb@hll.mpg.de

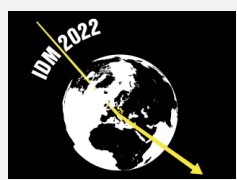

## Abstract

In the search for dark matter particle candidates, the mass region below 1 GeV/$c^2$ is relatively unprobed. Utilizing a low-noise silicon sensor as a sensitive target material, we aim to study the event signature of recoils between dark matter candidates and bound electrons. As the deposited energy is only a few eV, a sensor capable of detecting these low signals is required. We present first measurements on a prototype pixel matrix. It is based on the RNDR DePFET principle and provides a deep sub-electron readout noise of 0.2 e$^-$ and below.

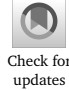

## 1 Introduction

Dark matter is one of the big mysteries of modern physics. While several independent observations indicate the existence of Dark Matter and its existence is undisputed, its nature is still unknown [1, 2]. A possible explanation for Dark Matter is the existence of a new particle, which is searched for by several experiment [3]. Historically, many experiments search for Dark Matter-nucleus scattering, however also the scattering of a Dark Matter particle on an electron is possible [4]. Using a silicon sensor, the interaction will happen with a bound electron of the silicon atom itself. So the silicon subtrate is at the same time target material and detector. Per interaction, only a few eV of energy is transferred to the electron resulting in

the generation of a few signal electrons inside the silicon. To resolve these small signals a near noiseless sensor is required [5]. The limit for conventional filtering techniques is the 1/f noise in the detector system [6]. Utilizing a Repetitive Non-Destructive Readout (RNDR) Depleted P-channel Field Effect Transistor (DePFET), multiple measurements of the same charge signal are possible, reducing the error of the mean following the central limit theorem. That way it is possible to detect and distinguish single electrons generated within the silicon volume. The long-term objective of Direct dArk matter detection using DEPFET with reptitive Non-destructive readout Application Experiment (DANAE) is to operate a large volume silicon detector (several grams of mass) over several years to access a non-excluded parameter space for dark matter detection [5]. The benefit and detection limits for different configurations of such a sensor have been discussed in [5]. In this paper we show first measurements from a 64x64 pixel RNDR DePFET matrix. The matrix achieved single electron resolution over most of its pixels. We investigated two of the factors limiting the resolution of this matrix and will briefly discuss their likely origins, possible remedies and future improvements.

## 2 Measurement setup

As discribed in [7], the measurement setup was designed, to test a $64 \times 64$ pixel RNDR DePFET matrix at temperatures as low as -200°C (93 K). The first tested RNDR DePFET sensor has a pixel size of $50 \times 50 \, \mu m^2$. The sensor was operated using a simple scheme, starting with a clear of all rows one by one. With the applied timings, this took $1.7 \, \mu s$ per row or about $109 \, \mu s$ for the complete sensor. This is followed by an adjustable illumination time. Next the collected signal is read out. For this, we implemented an $n$-fold trapezoidal weighting, with a multiplexing of the signal after every weighting function. Signal weighting and multiplexing took $18 \, \mu s$ and $6.4 \, \mu s$ respectively. That way we digitize $64 \times n \times 64$ data values per frame. In this context $n$ represents the number of repetitions of a readout.

## 3 Low intensity light illumination

To evaluate the single electron resolution of the matrix, the sensor was illuminated with a low light level using a pulsed led. This was done during a dedicated illumination time before the readout of the matrix.

### 3.1 Single electron resolution

Subtracting the offset – recorded with a set of dark frames – as well as the row wise common mode and summing up the resulting values from all pixels – without any bad pixel filtering – we got the histogram shown in figure 1. The zero as well as one, two and three electron peaks are clearly visible in the non-calibrated histogram. The shift of position between the spectra is caused by the offset calculation. The width of the peaks reduces with the number of repetitions as expected from the RNDR working principle. The width of the zero peaks is 0.71 ADU at 100 repetitions and 0.47 ADU at 800 repetitions, corresponding to $0.23 \, e^-$ and $0.14 \, e^-$ respectively. Examining the histograms closely, one can see that the valley between two peaks is not reducing as expected. This is due to charge arriving (section 3.2.1) as well as charge getting lost (section 3.2.2) during the readout process.

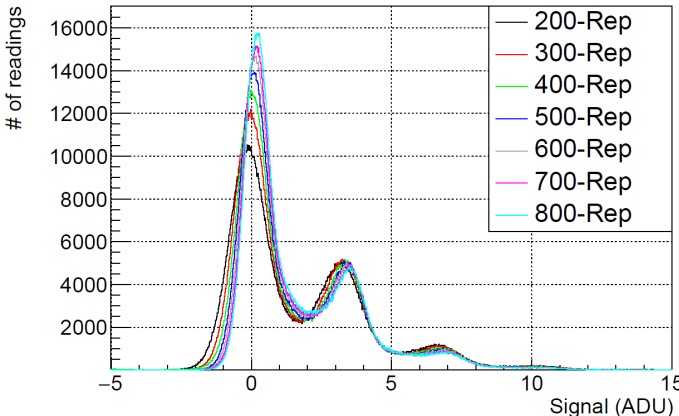

Figure 1: Histograms for different number of repetitions measured with the RNDR DePFET matrix. The width of the peaks reduces as expected from the RNDR principle. Only offset and common mode have been subtracted for each pixel.

## 3.2 Non constant charge

The signal of one pixel over 600 repetitions, is shown in figure 2a. The solid line shows a linear fit to the datapoints. The slope of the linear fit can be interpreted as the change of charge over the readout cycle. If the transfer of charge was lossless and generation of the charge during the readout negligible, we expect the slopes of different events to follow a Gaussian distribution centered around 0. We have selected events from pixels that showed a valid gain calibration, with an event being an averaged signal, exceeding the noise by $2\,\sigma$. The slopes for these events were fitted and histogrammized (figure 2b). This histogram was then fitted with a Gaussion (figure 2b red). As shown, the centroid of the Gaussian is close to zero and most events have a slope within the distribution as expected. However, both a wing on the positive as well as the negative side are visible. These indicates both, charge generation within the device as well as charge losses during the transfer of charge between the two DePFETs of one pixel.

### 3.2.1 Charge generation

In addition to the signal, charge can be generated by thermal excitation, leading to leakage current and during the switching of the DePFETs gate by impact ionization – similar to the effect known as spurious charge generation in CCDs [8]. The leakage current can be reduced by cooling the sensor. Impact ionization on the other hand increases with decreasing temperature as the mean free path length increases. Further studies to distinguish these two components and reduce impact ionization in DePFETs are in progress.

### 3.2.2 Charge losses

Charge losses might result from a of the signal charge close to the interface of the transfer gate if the transfer time is long enough for a trap at the interface to capture a signal electron. The technology of the current prototypes is optimized for X-ray spectroscopy, manufactured with a low dose blanked deep-n implant. This implant serves as a transfer channel for the RNDR DePFET. With the dose of the implant and the voltages applied to the gate and transfer gate during the transfer, the transfer channel is not optimally separated from the interface allowing for trapping of signal charge. Increasing the implant dose will establish the transfer channel further from the interface and prevent charge losses.

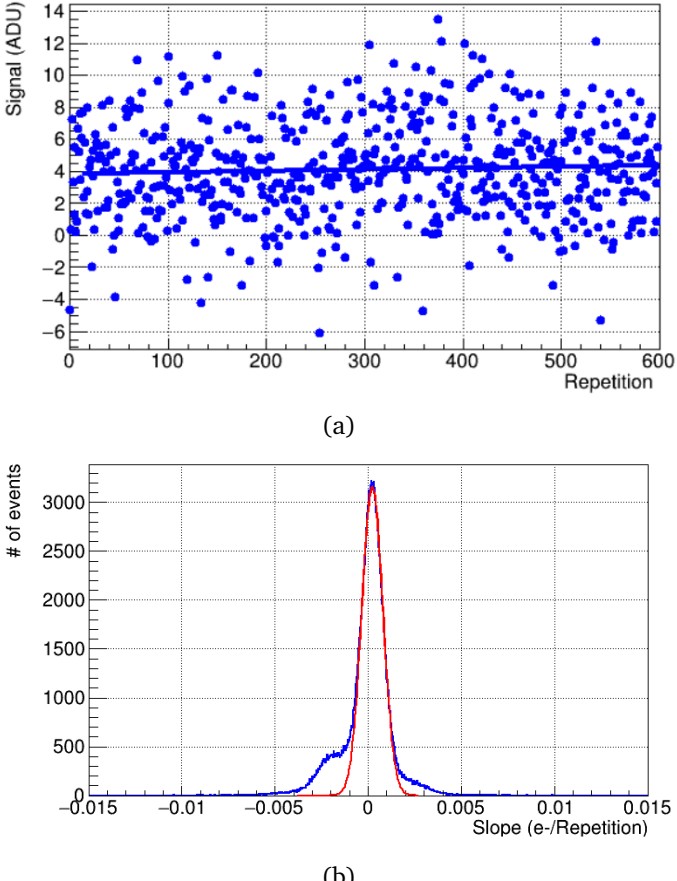

(a)

(b)

Figure 2: (a) Shown is the signal of one pixel over 600 repetitions. Fitting a linear curve to the data, the slope of the signal can be determined. (b) The slopes of the linear fits for different events from selected pixels is histogrammized. Most slopes are following a gaussian distribution with mean value close to zero. However, wings on both sides of the gaussian are present, indicating that the charge can increase or decrease during the readout.

### 3.2.3   Removing limiting events

By fitting a Gaussian to the distribution of slopes, we can distinguish events affected by generation or loss during the device readout. The fit is indicated in figure 2b in red. Using this fit we can define a filter that excludes events with a slope 2.5 $\sigma$ more negative as well as positive than the mean position of the Gaussian. The effect of this filter is indicated in figure 3. The green curve shows a histogram of the recorded events. By removing events that show charge losses (Neg. Filter) the valley between the one and two electron peaks improves significantly. The valley improves further if we also filter events that show a positive slope, i.e. charge collection during the readout process.

## 4   Summary and Outlook

We presented first measurements on a small prototype DePFET RNDR matrix of $64 \times 64$ pixels. As anticipated, the matrix can provide low readout noise of $0.23\,\mathrm{e}^-$ at 100 repetitions and $0.14\,\mathrm{e}^-$ at 800 repetitions, similar to skipper CCDs [9] but limited by charge generation as well

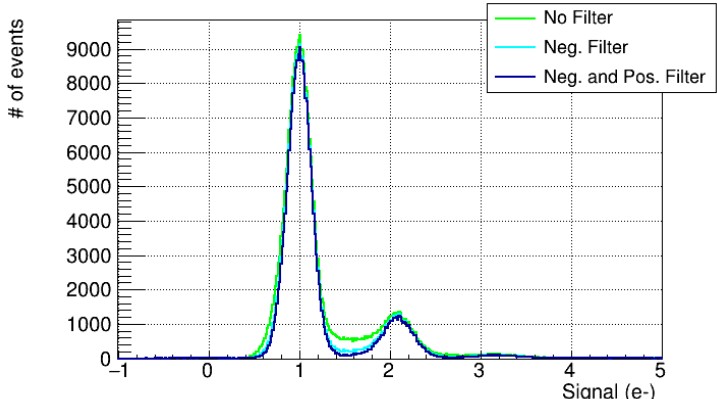

Figure 3: Histogram of the calibrated events. In *green* all events, in *cyan* events showing charge losses are excluded while in *blue* additionally events that show charge generation are excluded.

as charge losses for this prototype. At this stage we can detect these events and remove them from the data. The limitations will be mitigated by an optimized technology and operation conditions for a next generation of RNDR DePFETs. Further qualification of the current device will also include more advanced readout schemes with sparse clear only every few frames, providing an incremental readout of the collected charge. In the long term we plan to assemble a RNDR based sensor array with a mass of several grams of sensitive volume.

## Acknowledgements

**Funding information** This work was supported by the Austrian Science Fund (FWF) under the project ID M-2830.

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
