# Peer review of "First measurement Results from DANAE - Demonstrating DePFET RNDR on a prototype Matrix"

_SciPost Physics Proceedings, doi:SciPost Phys. Proc. 12, 066 (2023)_

## Round 1 · Referee Report · Anonymous (Referee 1) · 2022-11-28

Strengths

The paper mostly concentrate on the first measurements done to test the prototype. The text is well written.

Report

I recommend to apply some minor changes bellow.

Requested changes

  • “as sensitive target material “ -> “as a sensitive target material ”
  • “scattering of a Dark Matter particle on an electron is possible ” - please provide a ref.
  • “The long-term objective of DANAE is” - can you explain what is DANAE (abbreviation, experiment, project)?
  • “large volume silicon detector ” - dimensions will be useful here
  • “-200°C” - provide in brackets the corresponding value in K
  • “Next the collected signal is read out. For the readout …” - if it is possible avoid read out and readout close to each other.
  • Fig.2 provide more details for (b).
  • “We selected events ” - “We have selected events ”
  • “In addition to the signal, charge can be generated by thermal excitation, leading to leakage current and, furthermore, charge can be generated during the switching of the DePFETs gate by impact ionization – similar to the effect known as spurious charge generation in CCDs [7].” - “charge can be generated” second time is not necessary.
  • “This implant serves as transfer channel ” - > “This implant serves as a transfer channel ”
  • Fig. 3, I would use italic for colors.
  • “neglect events” ??
  • For the last section “summary” I would recommend to provide a comparison of achieved “low readout noise” to used by others. A few word on future plans would be interesting too.

  • validity: -
  • significance: -
  • originality: -
  • clarity: -
  • formatting: -
  • grammar: -

Author:  Alexander Bähr  on 2022-12-14  [id 3135]

(in reply to Report 1 on 2022-11-28)
Category:
question

Thanks for your suggestion. I implemented all, I'm only having difficulty to understand what you suggest with:
"For the last section “summary” I would recommend to provide a comparison of achieved “low readout noise” to used by others."
Could you elaborate? Thanks.

Anonymous on 2022-12-18  [id 3143]

(in reply to Alexander Bähr on 2022-12-14 [id 3135])

Dear author,

the intention of the comment was to give a reader understanding how "low"the level of readout noise in comparison with similar experiments, set-ups, devices etc.

Anonymous on 2022-12-20  [id 3163]

(in reply to Anonymous Comment on 2022-12-18 [id 3143])
Category:
answer to question
correction

Thanks for the clarification. I revised the Summary to read, referencing published performance of a skipper CCD:

We presented first measurements on a small prototype DePFET RNDR matrix of 64×64 pixels.
As anticipated, the matrix can provide low readout noise of 0.23e− at 100 repetitions and
0.14 e− at 800 repetitions, similar to skipper CCDs [9] but limited by charge generation as well
as charge losses for this prototype. At this stage we can detect these events and remove them
from the data. The limitations will be mitigated by an optimized technology and operation
conditions for a next generation of RNDR DePFETs. Further qualification of the current device
will also include more advanced readout schemes with sparse clear only every few frames,
providing an incremental readout of the collected charge. In the long term we plan to assemble
a RNDR based sensor array with a mass of several grams of sensitive volume.

Attachment:

IDM_paper.pdf

Anonymous on 2022-12-21  [id 3167]

(in reply to Anonymous Comment on 2022-12-20 [id 3163])
Category:
answer to question

Thank you. Please resubmit the new version using resubmission process.

---

## Round 2 · Author Response

• “as sensitive target material “ -> “as a sensitive target material ”

  • “scattering of a Dark Matter particle on an electron is possible ” - please provide a ref. included reference to " R. Essig, M. Fernandez-Serra, J. Mardon, A. Soto and T. Volansky, Direct detection of sub-gev dark matter with semiconductor targets, JHEP 05 (2015), doi:10.1007/JHEP05(2016)046."

  • “The long-term objective of DANAE is” - can you explain what is DANAE (abbreviation, experiment, project)? abbreviation included.

  • “large volume silicon detector ” - dimensions will be useful here included in the text as (several grams of sensitive mass). Details will be fixed in the future

  • “-200°C” - provide in brackets the corresponding value in K done

  • “Next the collected signal is read out. For the readout …” - if it is possible avoid read out and readout close to each other. Now reads: "Next the collected signal is read out. For this, we implemented an n-fold trapezoidal weighting,"

  • Fig.2 provide more details for (b). caption now reades: "The slopes of the linear fits for different events from selected pixels is histogrammized. Most slopes are following a gaussian distribution with mean value close to zero. However, wings on both sides of the gaussian are present, indicating that the charge can increase or decrease during the readout."

  • “We selected events ” - “We have selected events ”

  • “In addition to the signal, charge can be generated by thermal excitation, leading to leakage current and, furthermore, charge can be generated during the switching of the DePFETs gate by impact ionization – similar to the effect known as spurious charge generation in CCDs [7].” - “charge can be generated” second time is not necessary. removed repetition

  • “This implant serves as transfer channel ” - > “This implant serves as a transfer channel ”

  • Fig. 3, I would use italic for colors. done

  • “neglect events” ?? changed to "filter events"

  • For the last section “summary” I would recommend to provide a comparison of achieved “low readout noise” to used by others. A few word on future plans would be interesting too. changed paragraph to : We presented first measurements on a small prototype DePFET RNDR matrix of 64×64 pixels. As anticipated, the matrix can provide low readout noise of 0.23e− at 100 repetitions and 0.14 e− at 800 repetitions, similar to skipper CCDs [9] but limited by charge generation as well as charge losses for this prototype. At this stage we can detect these events and remove them from the data. The limitations will be mitigated by an optimized technology and operation conditions for a next generation of RNDR DePFETs. Further qualification of the current device will also include more advanced readout schemes with sparse clear only every few frames, providing an incremental readout of the collected charge. In the long term we plan to assemble a RNDR based sensor array with a mass of several grams of sensitive volume.

---

## Editorial Decision

published